# Identification and Characterization of Development-Related microRNAs in the Red Flour Beetle, *Tribolium castaneum*

**DOI:** 10.3390/ijms24076685

**Published:** 2023-04-03

**Authors:** Chengjun Li, Wei Wu, Jing Tang, Fan Feng, Peng Chen, Bin Li

**Affiliations:** Jiangsu Key Laboratory for Biodiversity and Biotechnology, College of Life Sciences, Nanjing Normal University, Nanjing 210023, China

**Keywords:** microRNAs, *Tribolium castaneum*, metamorphosis, wing development

## Abstract

MicroRNAs (miRNAs) play important roles in insect growth and development, but they were poorly studied in insects. In this study, a total of 883 miRNAs were detected from the early embryo (EE), late larva (LL), early pupa (EP), late pupa (LP), and early adult (EA) of *Tribolium castaneum* by microarray assay. Further analysis identified 179 differentially expressed unique miRNAs (DEmiRNAs) during these developmental stages. Of the DEmiRNAs, 102 DEmiRNAs exhibited stage-specific expression patterns during development, including 53 specifically highly expressed miRNAs and 20 lowly expressed miRNAs in EE, 19 highly expressed miRNAs in LL, 5 weakly expressed miRNAs in EP, and 5 abundantly expressed miRNAs in EA. These miRNAs were predicted to target 747, 265, 472, 234, and 121 genes, respectively. GO enrichment analysis indicates that the targets were enriched by protein phosphorylation, calcium ion binding, sequence-specific DNA binding transcription factor activity, and cytoplasm. An RNA interference-mediated knockdown of the DEmiRNAs tca-miR-6-3p, tca-miR-9a-3p, tca-miR-9d-3p, tca-miR-11-3p, and tca-miR-13a-3p led to defects in metamorphosis and wing development of *T. castaneum*. This study has completed the identification and characterization of development-related miRNAs in *T. castaneum*, and will enable us to investigate their roles in the growth and development of insect.

## 1. Introduction

microRNAs (miRNAs) are a type of endogenous non-coding small RNAs that can regulate post-transcriptional gene expression through complementary base-pairing with the messenger RNA (mRNA) of the target gene to form RNA-induced silencing complex [1]. Usually, primary miRNA is transcribed by RNA polymerase-II and then cleaved by the Drosha-DGCR8 complex to form a smaller structure called pre-miRNA [2]. This structure is transported to the cytoplasm by Exportin-5 and then enzymatically cleaved by Dicer to form mature miRNA [3,4]. miRNAs are associated with many biological processes, such as development, metabolism, apoptosis, autophagy, immunity and virus transmission [5,6,7,8,9,10].

Among insects, identification of miRNAs has been completed in representative insects, including *Drosophila melanogaster* [11,12], *Aedes albopictus*, *Culex quinquefasciatus* [13], *Bombyx mori* [14], *Tribolium castaneum* [15,16,17,18], *Apis mellifera* [19,20], *Acyrthosiphon pisum* [21] and *Pediculus humanus humanus* [22]. However, these studies pay little attention to development-related miRNAs at the transcriptome level. However, there have been many studies to explore functions of miRNAs in the insect. MiR-125-3p and miR-276b-3p target *orb2* to regulate spermatogenesis of *Bactrocera dorsalis* [23], whereas miR-311-3p and miR-309a are involved in ovarian development [24,25]. In *A. aegypti*, miR-275, miR-305 [26] and miR-2940/Clumsy are responsible for egg development [27]. Knockdown of tca-miR-8-3p in the late larva stage led to defects in embryogenesis and metamorphosis of *T. castaneum* [28]. Overexpression or knockdown of miR-2 blocked the transition of larva to pupa of *Blattella germanica* [29,30]. Overexpression of miR-927 in the *Drosophila* fat body significantly decreased the expression of *Kr-h1* and resulted in reduced oviposition, increased mortality, delayed pupation and reduced pupal size [31]. MiR-252-5p/Abelson interacting protein (Abi) was found to regulate the metamorphosis of larva-to-adult by controlling the cell cycle [9]. Accumulation of miR-2834 resulted in pupal and adult deformation and reduced fecundity in females of *B. mori* [32]. MiR-14 and miR-2766 regulate tyrosine hydroxylase to control larval-pupal metamorphosis in *Helicoverpa armigera* [33]. MiRNA let-7 is required for hormone regulation of metamorphosis in *B. mori* [34], and let-7-5p targets *Kr-h1* to regulate reproductive diapause in *Galeruca daurica* [35]. MiR-282-5p regulates larval moulting process by targeting chitinase 5 in *B. mori* [36]. Overexpression of miR-8 in the corpus allatum of *D. melanogaster* increased cell size of the gland and expression of *JHAMT*, resulting in pupal lethality [37]. These studies demonstrate that miRNAs are involved in embryonic development and metamorphosis. Moreover, miR-9B mediates dimorphism and development of wing in *A. pisum* [38], and *Drosophila* miR-958 regulates bone morphogenetic protein signaling to maintain tissue homeostasis [39]. The miR-125/Chinmo pathway regulates dietary restriction-dependent enhancement of lifespan in *Drosophila* [40], and miR-190 acts in the neurons to regulate lifespan, neuronal maintenance and age-related locomotor activity of male flies [41]. It is concluded that miRNAs play critical roles in tissue morphogenesis and lifespan of insect. Additionally, *Drosophila* miR-956 fine-tunes Notch signaling activity in intermediate, enteroblast progenitor cells to control enterocyte differentiation [42]. Clustered miRNAs miR-306 and miR-79 function as novel tumor-suppressors that specifically eliminate JNK-activated tumors in *Drosophila* [43]. These studies indicate that miRNAs participate in multiple developmental processes, and we need to reveal more development-related miRNAs at the genome level in insects.

To address these issues, we identified differentially expressed unique miRNAs (DEmiRNAs) among five developmental stages by microarray assay, predicted the targets for these DEmiRNAs, functionally annotated these targets by Gene ontology (GO) analysis, and investigated the roles of five DEmiRNAs in development by RNA interference. This study will promote the understanding of miRNAs regulating growth and development in insects.

## 2. Results

### 2.1. Identification of DEmiRNAs among Five Developmental Stages

A total of 3638 unique miRNAs of *T. castaneum* were synthesized as microfluidic chips to screen differentially expressed miRNAs (DEmiRNAs) among five stages, including early embryo (EE), late larva (LL), early pupa (EP), late pupa (LP) and early adult (EA). After signal normalization and ANOVA analysis, microarray assay detected 883 miRNA sequences from five different developmental stages, including 423 differentially expressed miRNA (DE) sequences (*p* < 0.05) and 460 stably expressed miRNA (SEmiRNA) sequences. Of the DEmiRNA sequences, there are 228 DEmiRNA sequences with signal intensity ≥500 and 195 DEmiRNA sequences having the signal <500. Among the SEmiRNA sequences, 57 miRNA sequences had the signal strength ≥500 while 403 miRNA sequences possessed the signal intensity <500 (Figure 1). After removing the redundant sequences, 179 differentially expressed unique miRNAs (DEmiRNAs) with signal strength ≥100 in at least one developmental stage were kept for further analysis (Appendix A).

To verify the results of microarray assay, quantitative real-time PCR (RT-PCR) was used to examine the expression of 15 DEmiRNAs during development. The expression profiles of tca-miR-184-3p (Figure 2A), tca-miR-276-3p (Figure 2B), tca-miR-263a-5p (Figure 2C), tca-miR-281-5p (Figure 2D), PC-3p-1361-938 (Figure 2E), PC-3p-37091-2 (Figure 2F), PC-5p-538866-1 (Figure 2G), PC-5p-136197-6 (Figure 2H), tca-miR-1-3p (Figure 2I), tca-miR-8-3p (Figure 2J), tca-miR-309a-3p (Figure 2K), tca-miR-750-3p (Figure 2L), tca-let-7-5p (Figure 2M), tca-miR-3869-3p (Figure 2N), and tca-miR-3834-5p (Figure 2O) by qRT-PCR were similar to those of microarray assay, demonstrating that the identification of DEmiRNAs by microarray assay is reliable.

### 2.2. Identification of DEmiRNAs between Two Developmental Stages

Since miRNAs from embryos, larvae, pupae, and adults have been obtained, it is necessary to identify DEmiRNAs between the two developmental stages. By microarray analysis, we found 142 DEmiRNAs (97 upregulated miRNAs and 45 downregulated miRNAs) between LL and EE, 140 DEmiRNAs (47 and 93) between EP and LL, 76 DEmiRNAs (50 and 26) between LP and EP, and 11 DEmiRNAs (7 and 4) between EA and LP (Appendix A). From the volcano plots of sequences with strong signal (≥500), 73, 84, 47, and 4 DEmiRNA sequences were found in pairs of LL-VS-EE, EP-VS-LL, LP-VS-EP, and EA-VS-LP, respectively (*p* < 0.05). Of these, 54 miRNA sequences were upregulated whereas 19 were downregulated during EE to LL development. Thirty-eight miRNAs were upregulated while 46 were downregulated when LL molt into EP. Thirty-three miRNAs were upregulated whereas 14 were downregulated during EP to LP development. One miRNA was upregulated and 3 miRNAs were downregulated when LP entered to EA (Figure 3).

### 2.3. Analysis of Stage-Specific DEmiRNAs

In order to explore the role of miRNAs in specific developmental stage, we next conducted the identification of stage-specific miRNAs. According to the expression patterns, 102 DEmiRNAs with the signal strength ≥500 were classified into five clusters (Figure 4A–E). The cluster A includes 53 embryo-specific highly expressed miRNAs, the cluster B comprises 20 embryo-specific lowly expressed miRNAs, the cluster C contains 19 miRNAs specifically highly expressed in LL, the cluster D consists of 5 miRNAs with specifically weak transcription in EP, and the cluster E has 5 miRNAs specifically highly expressed in EA (Table 1).

### 2.4. Target Prediction of DEmiRNAs

MiRNAs usually bind to targets to participate in multiple biological processes, and we subsequently predicted the targets of these DEmiRNAs by miRanda, PicTar, and TargetScan, respectively. A total of 2025 target genes with 18,734 target sites, 2953 target genes with 30,009 target sites, and 2205 target genes with 24,204 target sites were screened from miRanda, PicTar, and TargetScan, respectively. There are 68.4% common spots between miRanda and PicTar, 63% common spots between miRanda and TargetScan, and 74.4% common spots between PicTar and TargetScan. All three softwares shared 63% common spots. Totally 1863 common target genes were used for further analysis (Figure 5). For the DEmiRNAs, 747, 265, 472, 234, and 121 target genes were obtained from EE-specific highly expressed miRNAs (cluster A) and lowly expressed miRNAs (cluster B), LL-specific highly expressed miRNAs (cluster C), EP-specific lowly expressed miRNAs (cluster D), and EA-specific highly expressed miRNAs (cluster E), respectively (Table 1).

### 2.5. Functional Analysis of DEmiRNA Targets

To understand the function of these miRNAs, GO analysis was performed on their targets. The top six GO terms that enriched cluster A miRNA targets are protein phosphorylation (21 targets), ATPase activity (14 targets), multicellular organismal development (10 targets), hydrolase activity (9 targets), endosome (7 targets), and nuclear envelope (7 targets) (Figure 6A). The most enriched GO terms for cluster B are calcium ion binding (32), microsome (7), brain development (6), and cell-cell adhesion (6) (Figure 6B). The cluster C miRNA targets were significantly enriched in sequence-specific DNA binding transcription factor activity (19), regulation transcription, DNA dependent (17), membrane (17), and ion transport (8) (Figure 7A). The cluster D targets were enriched by cytoplasm (32), plasma membrane (20), signal transduction (15), and sequence-specific DNA binding transcription factor activity (14) (Figure 7B). The most enriched GO terms are RNA polymerase II distal enhancer sequence-specific DNA binding transcription factor activity (3) and electron carrier activity (3) for cluster E targets (Appendix A).

### 2.6. DEmiRNAs Are Required for Metamorphosis and Wing Development

Among these miRNAs, we selected five miRNAs (three from cluster A, and one each of cluster C and E) that have been revealed by GO analysis to play critical roles in development of the insect to verify the relationship between these miRNAs and development. Antagomirs specific for tca-miR-6-3p (Anta-6), tca-miR-9a-3p (Anta-9a), tca-miR-9d-3p (Anta-9d), tca-miR-11-3p (Anta-11), and tca-miR-13a-3p (Anta-13a) were injected into late larvae of *T. castaneum*, respectively. Results of the PCR analyses revealed that the mRNA levels of these miRNAs significantly decreased 48 h post-injection (Figure 8A,B). When the larva moults into a pupa, a knockdown of tca-miR-6-3p or tca-miR-13a-3p caused 9.56% or 16.08% of larvae to have defects during pupation of *T. castaneum*, and these pupae with defective wings were trapped in their old larval cuticle, but 23.51% of larvae injected with tca-miR-9d-3p-specific antagomirs became pupae with abnormal wings (Figure 8C). During the pupa to adult transition, silencing tca-miR-6-3p made 23.23% of insects moult into adults with the old pupal cuticle adhering to the adult body end, 24.17% of tca-miR-9a-3p knockdown adults had impaired wings, and 45.83%, 21.52% and 19.79% of insects separately treated by antagomirs of tca-miR-9d-3p, tca-miR-11-3p and tca-miR-13a-3p failed to initiate the ecdysis and arrested development (Figure 8C). These results suggest that the DEmiRNAs play essential roles in metamorphosis and wing development of *T. castaneum*.

## 3. Discussion

In this study, we identified DEmiRNAs among five developmental stages by microarray, predicted their targets, functionally annotated them by GO analysis of their targets, and investigated the developmental roles of five DEmiRNAs in *T. castaneum*.

In the past years, a number of miRNAs have been identified from *T. castaneum* [15,18]. By comparison with other developmental stages, we found 53 specially upregulated miRNAs (cluster A) and 20 downregulated miRNAs (cluster B) in the early embryonic (EE) stage of *T. castaneum* (Table 1, Figure 4A,B). Among the upregulated miRNAs, miR-7 exerts a role in the switch from the endocycle to gene amplification through its regulation of Tramtrack69 in *Drosophila* follicle cells [44], miR-9 specifically controls the expression of mesodermal genes to canalize myotendinous junction morphogenesis during embryogenesis [45], and miR-309 targets SIX4 and controls ovarian development in *A. aegypti* [46]. Of the downregulated miRNAs, let-7 could bind to the *Kr-h1* coding sequence to block oocyte maturation in *Locusta migratoria* [47], miR-1-3p is an early embryonic male sex-determining factor in the Oriental fruit fly *B. dorsalis* [48], miR14 regulates egg-laying by targeting EcR in *A. mellifera* [49], loss of miR-184 in *Drosophila* led to multiple severe defects during oogenesis and early embryogenesis, culminating in the complete loss of egg production [50], and miR-276 promotes egg-hatching synchrony by upregulating brahma in locusts [51]. These studies indicate that specifically high or low expression of miRNAs in EE is associated with embryogenesis of *T. castaneum*.

This study discovered 19 miRNAs specifically highly expressed in the LL of *T. castaneum* (cluster C) (Table 1, Figure 4C). Of these miRNAs, an overexpression of miR-8 in the whole body at the end of *Drosophila* larval development inhibited ecdysone biosynthesis and caused defects in metamorphosis and survival [52], and its overexpression in the corpus allatum increased cell size of the gland and expression of *JHAMT*, leading to pupal lethality [37]. Specific inhibition of miR-8-3p in *T. castaneum* late larvae induced metamorphosis defects in the development of wings, eyes, legs and embryos [28]. The ablation of *B. mori* miR-34 by transgenic CRISPR/Cas9 modulated ecdysone signaling and resulted in a severe developmental delay during the larval stage [53], and miR-281 regulates the expression of *EcR* isoform B in *B. mori* [54]. The injection of tca-miR-6-3p antagomirs caused defects in ecdysis during pupation and eclosion and wing development (Figure 8). Therefore, these miRNAs were believed to have critical roles in metamorphosis of *T. castaneum*.

The cluster D has 5 miRNAs specifically weakly expressed in EP (Table 1, Figure 4D), and they were predicted to target 234 protein-coding genes (Table 1). The GO enrichment analysis showed that these DEmiRNAs mainly participate in the cytoplasm (32), plasma membrane (20), signal transduction (15), and sequence-specific DNA binding transcription factor activity (14) (Figure 7B). In the cluster, miR-9a targeting senseless ensures the precise specification of sensory organ precursors in *Drosophila* [55], and it also target the *Drosophila* transcriptional regulator LIM-only to prevent apoptosis during wing development [56]; miR-8 null flies are smaller in size and defective in insulin signaling in fat body [57], gain and loss of miR-8 provoked developmental defects reminiscent of impaired Notch signaling [58], and it also acts as a negative regulator of Wnt signaling to affect eye and wing development in *D. melanogaster* [59]; miR-252 has vital roles in growth, development and cell cycle of insect [9,60,61]; miR-3017 contributes to larval to pupal to adult metamorphosis by targeting sarco/endoplasmic reticulum Ca^2+^ ATPase in *T. castaneum* [62]. Mutation of the *Drosophila* miR-6 cluster, which contains three miR-6 genes, showed a modest reduction in viability to adulthood [63]. A knockdown of tca-miR-6-3p also led to defects during the transition of larva to pupa to adult (Figure 8). These results suggest that the cluster D miRNAs have multiple functions in organ growth, development, and cell cycle.

The cluster E contains 5 EA-specific highly expressed miRNAs (Table 1, Figure 4E). Among these DEmiRNAs, the dietary of miR-316 antagomirs led to significantly higher mortality (>70%) and a lower proportion of winged morphs [64], miR-9 functions in muscle contraction [65], dendrite growth of sensory neurons [66], and wing development (Figure 8) [67] of insect, and miR-375 has important roles in forming short-term memory [68], circadian rhythms and sleep [69] of *D. melanogaster*. Therefore, these miRNAs participate in many aspects of adult development in insects.

In conclusion, we identified 179 DEmiRNAs among five developmental stages of *T. castaneum*, and found 102 DEmiRNAs specifically expressed in EE, LL, EP, and EA. GO analysis indicated these miRNAs played vital roles in development. A knockdown of five DEmiRNAs caused defects in metamorphosis and wing development. This study completed the identification of development-related miRNAs in *T. castaneum*.

## 4. Materials and Methods

### 4.1. Insect Strains

The Georgia-1 (GA-1) strain of *T. castaneum* was reared at 30 °C in 5% yeast flour [70]. They were used in all experiments of this study.

### 4.2. RNA Isolation and cDNA Synthesis

The total RNA was extracted from early embryos (1-day embryos), late larvae (20-day larvae), early pupae (1-day pupae), late pupae (6-day pupae), and early adults (1-day adults) using a Total RNA Purification Kit (LC Sciences, Houston, TX, USA) according to the manufacturer’s protocol, respectively. The quantity and purity of the total RNA were analyzed with the Bioanalyzer 2100 by RNA 6000 Nano LabChip Kit (Agilent, Houston, TX, USA) with RIN number >7.0. In addition, 500 ng of total RNA was converted to cDNA using HiScript^®^ III RT SuperMix for qPCR (+gDNA wiper) (Vazyme, Nanjing, China).

### 4.3. Microarray Assay

A poly (A) tail was added to the 3′ end of 4 µg total RNAs of each sample by the poly (A) polymerase. An oligonucleotide tag was then ligated to the poly (A) tail for subsequent fluorescent dye staining. Hybridization was performed overnight on a µParaflo microfluidic chip by a micro-circulation pump (Atactic Technologies, Texas, TX, USA) [71]. On the microfluidic chip, each detection probe consisted of a chemically modified nucleotide coding segment that was complementary to the target miRNA and a spacer segment of polyethylene glycol to extend the coding segment away from the substrate. The detection probes were made in situ synthesis using photogenerated reagent (PGR) chemistry. The hybridization melting temperatures were balanced by chemical modifications of the detection probes. Hybridization used 100 L 6 × SSPF buffer (0.90 M NaCl, 60 mM Na_2_HPO_4_, 6 mM EDTA, pH 6.8) containing 25% formamide at 34 °C. After RNA hybridization, tag-conjugating Cy3 dye was circulated through the microfluidic chip for dye staining. Fluorescence images were collected using a laser scanner (GenePix 4000B, Molecular Device, California, CA, USA) and digitized using Array-Pro image analysis software (Media Cybernetics, Maryland, MD, USA). Data were analysed by first subtracting the background and then normalizing the signals using a LOWESS filter (Locally-weighted Regression, California, CA, USA) [72]. Each microarray contained 16 groups of 5sRNA probes as control, and these controls of signal would >10 to insured small RNAs were existed in the detected samples.

### 4.4. Identification of Differentially Expressed miRNAs

Fluorescence images were collected using a laser scanner (GenePix 4000B, Molecular Device) after RNA hybridization. The “Array-Pro Analyzer” (Media Cybernetics) was used for image digitization and background subtraction [71]. Normalization was carried out using the locally weighted scatterplot smoothing (LOWESS) method on the subtracted background data [72]. Differentially expressed miRNAs (DEmiRNAs) between developmental stages were filtered using ANOVA and t-tests [73].

### 4.5. Target Prediction for DEmiRNAs

Four different algorithms, including NCBI blast, PicTar (http://pictar.mdc-berlin.de/, accessed on 20 May 2022), miRanda (http://mirdb.org/miRDB/, accessed on 20 May 2022), and TargetScan (http://www.targetscan.org/vert_61/, 20 May 2022), were used to predict the potential targets of DEmiRNAs. PicTar calculates the free energy of miRNA, which represents the binding capability to the target gene [74]. miRanda analyses the free energy of miRNA binding capability and the binding site between miRNA and target gene [75]. TargetScan searches for complementary matches (called the “seed match”) of miRNA 2-8 bases (numbered from the 5′ end) with the 3′UTR sequence of mRNA [76]. Only miRNA-target interactions predicted by all three algorithms were used for further analysis.

### 4.6. Enrichment Analyses of Predicted DEmiRNA Targets

We used Blast2GO [77] to annotate the gene ontology (GO) term for DEmiRNA targets. A GO term with false discovery rate (FDR) ≤ 0.01 was judged to be a significantly enriched one.

### 4.7. Quantitative Real-Time PCR

Total RNAs were separately collected from samples and then converted to cDNAs for each miRNA by RT-primer (Appendix A) using HiScript^®^ III RT SuperMix for qPCR (+gDNA wiper) (Vazyme, Nanjing, China). The quantitative real time-PCR (qRT-PCR) was performed using the Hieff q-PCR Green Master Mix (Yeasen, Shanghai, China) to examine the expression of miRNAs. The relative levels of miRNAs were normalized to the level of U6 snRNA using the ^ΔΔ^CT method [78]. The results were analyzed from three independent samples. The primer sequences for these miRNAs are listed in Appendix A.

### 4.8. Antagomirs Synthesis and Injection

Antagomirs were obtained from GenePharma (Shanghai, China). Antagomirs were synthesized following its mature miRNA sequences: 5′-mA.*.mA.*.mA.mA.mA.mG.mA.mA.mC.mA.mG.mC.mC.mA.mC.mU.mG.mU.mG.*.mA.*. mU.*.mA.*. -Chol-3′ to miR-6-3p (Anta-6), 5′-mU.*.mA.*.mA.mC.mU.mC.mC.mG.mG.mU.mA.mA.mC.mC.mU.mA.mG.mC.mU.mU.*. mU.*.mA.*.mU.*. -Chol-3′ to miR-9a-3p (Anta-9a), 5′-mC.*.mU.*.mU.mU.mG.mG.mU.mG.mA.mA.mU.mC.mU.mA.mG.mC.mU.mU.*.mU.*.mA.*. mU.*. -Chol-3′ to miR-9d-3p (Anta-9d), 5′-mA.*.mG.*.mC.mU.mA.mG.mA.mA.mC.mU.mC.mU.mG.mC.mC.mU.mG.mU.mG.*.mA.*. mU.*.mG.*. -Chol-3′ to miR-11-3p (Anta-11), 5′-mA.*.mG.*.mC.mU.mC.mA.mU.mC.mA.mA.mA.mG.mU.mG.mG.mC.mU.mG.mU.mG.*. mA.*.mU.*.mA.*. -Chol- 3′ to miR-13a-3p (Anta-13a). Here, “*” is a PS backbone instead of the usual PO backbone. “m” is an OCH3 group on the 2′ end of the base instead of the usual OH group, and there is 3′cholesterol group added to each RNA oligo for potency reasons [79]. The Anta-8s and controls were injected into late larvae at a dose of 200 μM in a volume of 200 nL. Sequence of 5′-ACGUGACACGUUCGGAGAATT-3′ was served as stable negative control. Approximately 25 to 30 individuals per group were injected with antagomirs, and the injection was repeated for more than three biological samples. Three individuals were used to detect the knockdown efficiency of miRNAs at 48 h post antagomirs injection by qRT-PCR.

### 4.9. Statistical Analysis

Data were statistically analyzed by Student’s *t*-test and analysis of variance (ANOVA) using the graphic software Prism (Graph Pad Software, v8.2.1, San Diego, CA, USA). All data are presented as the mean ± standard error of the mean (SEM).

## Figures and Tables

**Figure 1 ijms-24-06685-f001:**
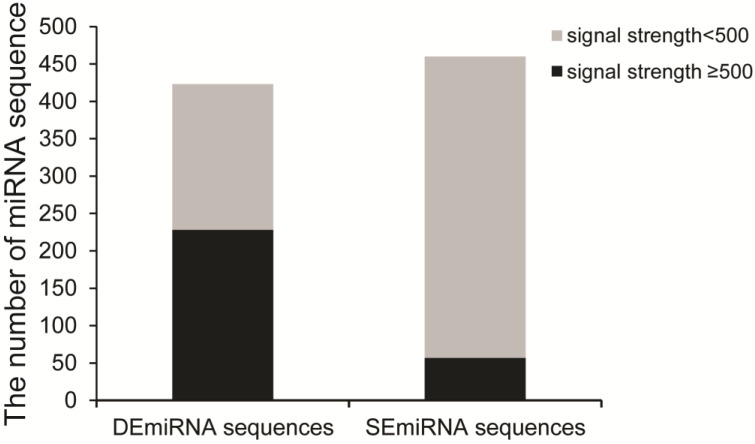
Identification of DEmiRNAs among five developmental stages. The grey column indicates miRNA with signal strength <500, while the black column represents miRNA with signal strength ≥500. Three biological replications were performed.

**Figure 2 ijms-24-06685-f002:**
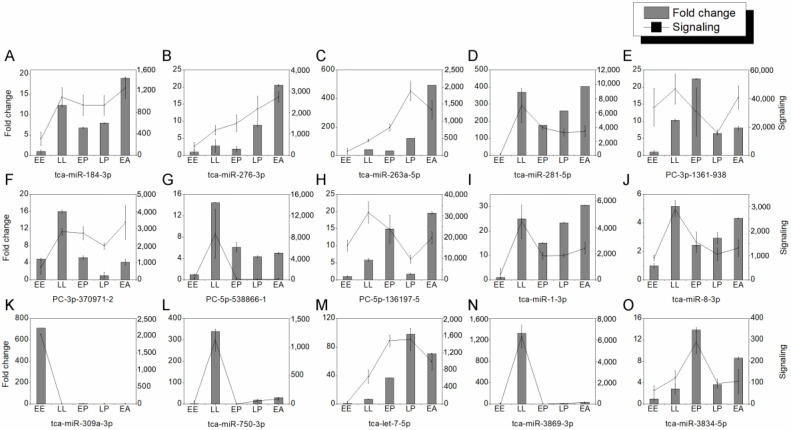
The expression profiles of 15 miRNAs (**A**–**O**) during development by qRT-PCR. Three biological samples were used for qRT-PCR analysis, and the U6 snRNA was served as an internal reference.

**Figure 3 ijms-24-06685-f003:**
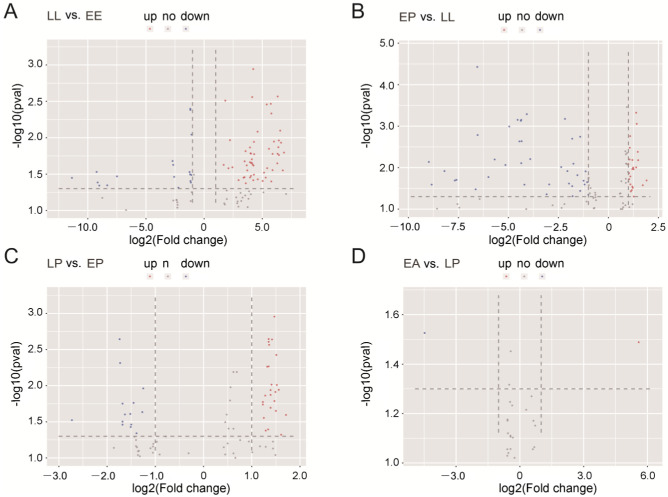
Volcano plots of miRNAs with signal strength ≥500 in the pairwise LL vs. EE (**A**), EP vs. LL (**B**), LP vs. EP (**C**), and EA vs. LP (**D**). Red points mean upregulated miRNAs (*p* < 0.05), blue points mean downregulated miRNAs (*p* < 0.05), and grey dots indicate miRNAs with no difference in expression between two developmental stages.

**Figure 4 ijms-24-06685-f004:**
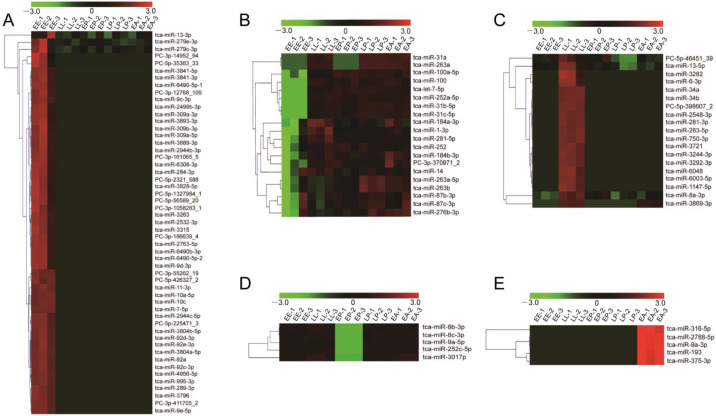
Hierarchical cluster of DEmiRNAs expression among five developmental stages. (**A**) The expression profiles of embryo specifically highly expressed miRNAs during development. (**B**) The expression patterns of embryo specifically lowly expressed miRNAs during development. (**C**) The expression patterns of miRNAs that specifically highly expressed in LL during development. (**D**) The expression patterns of EP specifically lowly expressed miRNAs during development. (**E**) The expression profile of EA specifically highly expressed miRNAs among five stages. Red indicates highly expressed miRNAs, and green indicates lowly expressed miRNAs. Three biological replications were analzsed.

**Figure 5 ijms-24-06685-f005:**
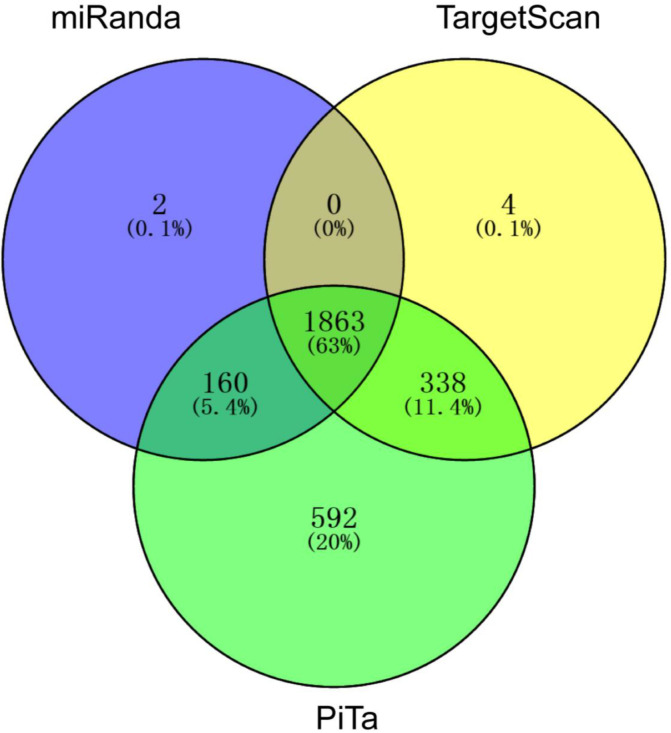
Venn diagram of target prediction of DEmiRNAs. Target genes of 179 DEmiRNAs were predicted by three programs: PicTar (http://pictar.mdc-berlin.de/, accessed on 20 May 2022), miRanda (http://mirdb.org/miRDB/, accessed on 20 May 2022), and TargetScan (http://www.targetscan.org/vert_61/, accessed on 20 May 2022).

**Figure 6 ijms-24-06685-f006:**
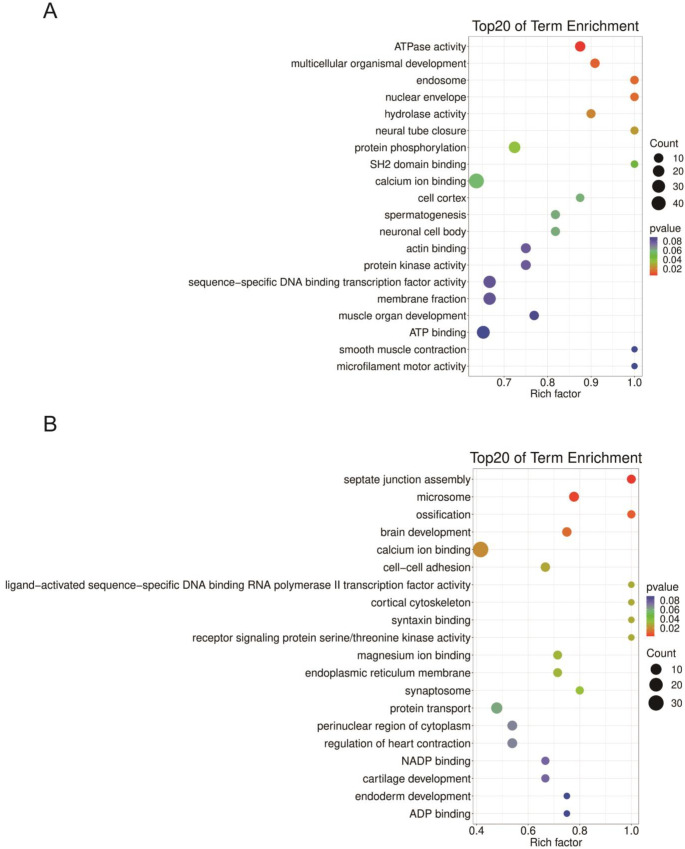
GO enrichment analysis of cluster A (**A**) and cluster B (**B**) DEmiRNA targets. The *Y*-axis indicates the GO term, and the *X*-axis represents the rich factor.

**Figure 7 ijms-24-06685-f007:**
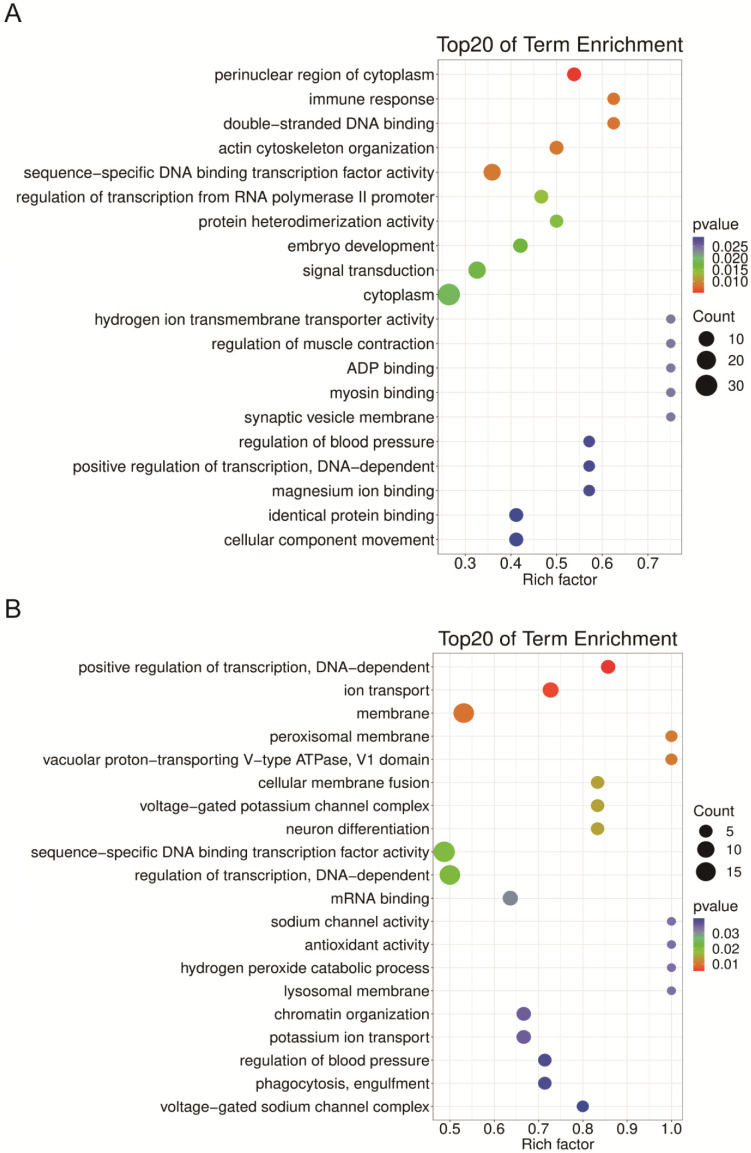
GO enrichment analysis of cluster C (**A**) and cluster D (**B**) DEmiRNA targets.

**Figure 8 ijms-24-06685-f008:**
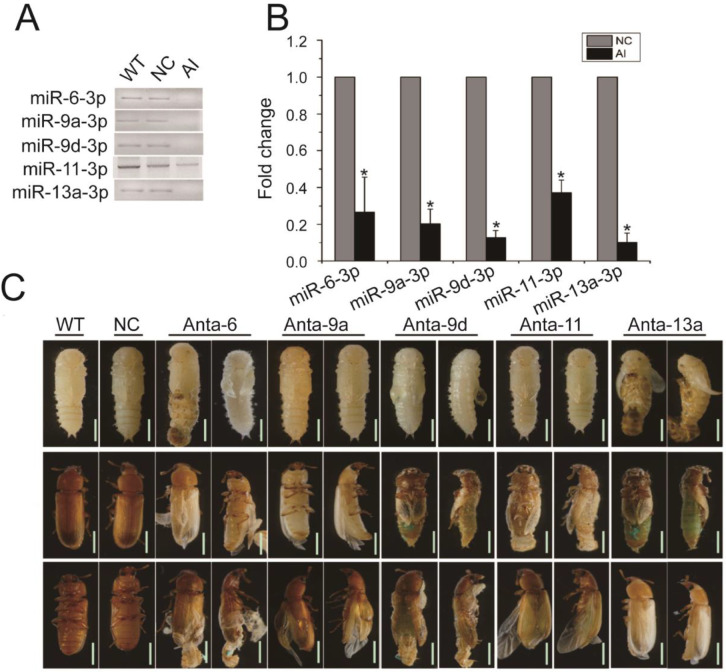
Effects of miR-6-3p, miR-9a-3p, miR-9d-3p, miR-11-3p, and miR-13a-3p on development of *T. castaneum*. (**A**,**B**) The expression level of miR-6-3p, miR-9a-3p, miR-9d-3p, miR-11-3p, and miR-13a-3p in wild-type (WT) insects, negative controls (NC) and the corresponding miRNA-specific antagomirs treated-samples (AI) at 48 h post injection by PCR. (**C**) Phenotypes induced by injecting miRNA-specific antagomirs. Three individuals from each group were used to determine gene expression level by PCR, and the results were calculated from three biological samples. “*” indicates *p* < 0.05. Scale bar: 1000 μm.

**Table 1 ijms-24-06685-t001:** Information for stage-specific DEmiRNAs during development.

Cluster	DEmiRNA No.	DEmiRNA	Target Sites	Target Genes
A	53	tca-miR-13-3p, tca-miR-279e-3p, tca-miR-279c-3p, PC-3p-14952_94, PC-5p-35363_33, tca-miR-3841-5p, tca-miR-3841-3p, tca-miR-6490-5p-1, PC-3p-12768_109, tca-miR-9c-3p, tca-miR-2499b-3p, tca-miR-309a-3p, tca-miR-3893-3p, tca-miR-309b-3p, tca-miR-309a-5p, tca-miR-3889-3p, tca-miR-2944b-3p, PC-3p-161065_5, tca-miR-6308-3p, tca-miR-284-3p, PC-5p-2321_588, tca-miR-3828-5p, PC-5p-1327984_1, PC-5p-56589_20, PC-3p-1058283_1, tca-miR-3263, tca-miR-2532-3p, tca-miR-3315, PC-3p-186639_4, tca-miR-2763-5p, tca-miR-6490b-3p, tca-miR-6490-5p-2, tca-miR-9d-3p, PC-3p-55262_19, PC-5p-426327_2, tca-miR-11-3p, tca-miR-10a-5p, tca-miR-10c, tca-miR-7-5p, tca-miR-2944c-5p, PC-5p-225471_3, tca-miR-3804b-5p, tca-miR-92d-3p, tca-miR-92e-3p, tca-miR-3804a-5p, tca-miR-92a, tca-miR-92c-3p, tca-miR-4956-5p, tca-miR-995-3p, tca-miR-289-3p, tca-miR-3796, PC-3p-411705_2, tca-miR-9e-5p	1792	748
B	20	tca-miR-31a, tca-miR-263a, tca-miR-100a-5p, tca-miR-100, tca-let-7-5p, tca-miR-252a-5p, tca-miR-31b-5p, tca-miR-31c-5p, tca-miR-184a-3p, tca-miR-1-3p, tca-miR-281-5p, tca-miR-252, tca-miR-184b-3p, PC-3p-370971_2, tca-miR-14, tca-miR-263a-5p, tca-miR-263b, tca-miR-87b-3p, tca-miR-87c-3p, tca-miR-276b-3p	728	379
C	19	PC-5p-46451_39, tca-miR-13-5p, tca-miR-3282, tca-miR-6-3p, tca-miR-34a, tca-miR-34b, PC-5p-398607_2, tca-miR-2548-3p, tca-miR-281-3p, tca-miR-283-5p, tca-miR-750-3p, tca-miR-3721, tca-miR-3244-3p, tca-miR-3292-3p, tca-miR-6048, tca-miR-6003-5p, tca-miR-1147-5p, tca-miR-8a-3p, tca-miR-3869-3p	665	460
D	5	tca-miR-8b-3p, tca-miR-8c-3p, tca-miR-9a-5p, tca-miR-252c-5p, tca-miR-3017b	389	234
E	5	tca-miR-316-5p, tca-miR-2788-5p, tca-miR-9a-3p, tca-miR-193, tca-miR-375-3p	148	132

## Data Availability

Data are contained within the articles or Appendix A.

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
