# Peer review of "Identification and Characterization of Development-Related microRNAs in the Red Flour Beetle, Tribolium castaneum"

_ijms, 2023, doi:10.3390/ijms24076685_

Round 1

Reviewer 1 Report

In this study, Li et al. aim to identify the mRNAs in the red flour beetle, Tribolium castaneum. Using a microarray approach, the authors identify a total of 883 miRNAs expresses along the different developmental stages, from embryo to early adult. Interestingly 179 out of those 883 mRNA presents a specific stage expression suggesting a possible role of these mRNA during development. Surprisingly, and contrary to what claims the title of the article, the authors do not further characterize the mRNAs identified, making the study incomplete. Although the experiments presented are well conduced, in my opinion the conclusions are premature and required additional experimental validation, as the “characterization” is based exclusively on target prediction. In summary, although the data presented in this study has some interest, is preliminary, and without a proper functional validation and characterization of the representative stage specific mRNAs, is not suitable for a journal like Int. J. Mol. Sci.

Author Response

In this study, Li et al. aim to identify the mRNAs in the red flour beetle, Tribolium castaneum. Using a microarray approach, the authors identify a total of 883 miRNAs expresses along the different developmental stages, from embryo to early adult. Interestingly 179 out of those 883 mRNA presents a specific stage expression suggesting a possible role of these mRNA during development. Surprisingly, and contrary to what claims the title of the article, the authors do not further characterize the mRNAs identified, making the study incomplete. Although the experiments presented are well conduced, in my opinion the conclusions are premature and required additional experimental validation, as the “characterization” is based exclusively on target prediction. In summary, although the data presented in this study has some interest, is preliminary, and without a proper functional validation and characterization of the representative stage specific mRNAs, is not suitable for a journal like Int. J. Mol. Sci.

Response: We accept the criticism, and investigated the roles of tca-miR-6-3p, tca-miR-9a-3p, tca-miR-9d-3p, tca-miR-11-3p, and tca-miR-13a-3p by injecting their specific antagomirs. Their functions were described as follows “Antagomirs specific for tca-miR-6-3p (Anta-6), tca-miR-9a-3p (Anta-9a), tca-miR-9d-3p (Anta-9d), tca-miR-11-3p (Anta-11), and tca-miR-13a-3p (Anta-13a) were injected into late larvae of T. castaneum, respectively. Results of PCR analyses revealed that the mRNA levels of these miRNAs significantly decreased 48 hours post-injection (Figure 8AB). During larva to pupa transition, knockdown of tca-miR-6-3p or tca-miR-13a-3p caused 9.56% or 16.08% of larvae to have defects during pupation, and these pupae with defective wings were trapped in their old larval cuticle, but 23.51% of larvae injected with tca-miR-9d-3p-specific antagomirs became pupae with abnormal wings (Figure 8C). During pupa to adult transition, of silencing tca-miR-6-3p made 23.23% of insects moult into adults with the old pupal cuticle adhering to the adult body end, 24.17% of tca-miR-9a-3p knockdown adults had impaired wings, and 45.83%, 21.52% and 19.79% of insects separately treated by antagomirs of tca-miR-9d-3p, tca-miR-11-3p and tca-miR-13a-3p failed to initiate the ecdysis and arrested development during eclosion (Figure 8C).” (lines 190-203) in the revised manuscript.

Reviewer 2 Report

In this study, a total of 883 miRNAs were detected from the early embryo (EE), late larva (LL), early pupa (EP), late pupa (LP) and early adult (EA) of Tribolium castaneum by microarray assay. Further analysis identified 179 differentially expressed unique miRNAs (DEmiRNAs) during development. Fron these DEmiRNAs, 102 exhibited stage-specific expression patterns during development, including 53 specifically highly expressed miRNAs and 20 lowly expressed miRNAs in EE, 19 highly expressed miRNAs in LL, 5 weakly expressed miRNAs in EP, and 5 abundantly expressed miRNAs in EA. Overall, the topic is interesting, but the results didn’t fully support the main conclusion. Moreover, the dataset is very limited for such studies. The authors have done just preliminary work in this field. Only this data is not enough to support the main conclusion. The authors must need to do RNA interference experiments to validate their results. These experiments must be done before submission of this manuscript to any SCI journal. Besides, the manuscript needs careful proofreading and revision. Grammar mistakes are undermining the significance of this study. Therefore, I suggest submitting it to the editing company for grammar modification. Due to all these major limitations, I cannot recommend it for publication in reputable journal like IJMS.

Besides, there are some major points:

- The introduction section is not coherent and very short. The authors must rewritten whole introduction section, add more detail information following latest published work in reputable journals. The information should be to the point.

- All figures should be optimize. Follow same color and font size of figures legends on x-axes and y-axes.

- The discussion section is very short. The authors must discuss their results in details according to latest work published in reputable journals.

Author Response

In this study, a total of 883 miRNAs were detected from the early embryo (EE), late larva (LL), early pupa (EP), late pupa (LP) and early adult (EA) of Tribolium castaneum by microarray assay. Further analysis identified 179 differentially expressed unique miRNAs (DEmiRNAs) during development. Fron these DEmiRNAs, 102 exhibited stage-specific expression patterns during development, including 53 specifically highly expressed miRNAs and 20 lowly expressed miRNAs in EE, 19 highly expressed miRNAs in LL, 5 weakly expressed miRNAs in EP, and 5 abundantly expressed miRNAs in EA. Overall, the topic is interesting, but the results didn’t fully support the main conclusion. Moreover, the dataset is very limited for such studies. The authors have done just preliminary work in this field. Only this data is not enough to support the main conclusion. The authors must need to do RNA interference experiments to validate their results. These experiments must be done before submission of this manuscript to any SCI journal. Besides, the manuscript needs careful proofreading and revision. Grammar mistakes are undermining the significance of this study. Therefore, I suggest submitting it to the editing company for grammar modification. Due to all these major limitations, I cannot recommend it for publication in reputable journal like IJMS.

Response: We accept the criticism, and investigated the roles of tca-miR-6-3p, tca-miR-9a-3p, tca-miR-9d-3p, tca-miR-11-3p, and tca-miR-13a-3p by injecting their specific antagomirs. Their functions were described as follows “Antagomirs specific for tca-miR-6-3p (Anta-6), tca-miR-9a-3p (Anta-9a), tca-miR-9d-3p (Anta-9d), tca-miR-11-3p (Anta-11), and tca-miR-13a-3p (Anta-13a) were injected into late larvae of T. castaneum, respectively. Results of PCR analyses revealed that the mRNA levels of these miRNAs significantly decreased 48 hours post-injection (Figure 8AB). During larva to pupa transition, knockdown of tca-miR-6-3p or tca-miR-13a-3p caused 9.56% or 16.08% of larvae to have defects during pupation, and these pupae with defective wings were trapped in their old larval cuticle, but 23.51% of larvae injected with tca-miR-9d-3p-specific antagomirs became pupae with abnormal wings (Figure 8C). During pupa to adult transition, of silencing tca-miR-6-3p made 23.23% of insects moult into adults with the old pupal cuticle adhering to the adult body end, 24.17% of tca-miR-9a-3p knockdown adults had impaired wings, and 45.83%, 21.52% and 19.79% of insects separately treated by antagomirs of tca-miR-9d-3p, tca-miR-11-3p and tca-miR-13a-3p failed to initiate the ecdysis and arrested development during eclosion (Figure 8C).” (lines 190-203) in the revised manuscript. We also edited the language throughout the manuscript.

Besides, there are some major points:

- The introduction section is not coherent and very short. The authors must rewritten whole introduction section, add more detail information following latest published work in reputable journals. The information should be to the point.

Response: We agree with the criticism, and added many latest articles into the section of introduction of revised manuscript.

- All figures should be optimize. Follow same color and font size of figures legends on x-axes and y-axes.

Response: We made change as suggested in the revised manuscript.

- The discussion section is very short. The authors must discuss their results in details according to latest work published in reputable journals.

Response: We accept the criticism, and added more contents into the section of discussion of the revised manuscript.

Reviewer 3 Report

In this work, the authors proposed an interesting study ‘” Identification and characterization of development related mi-croRNAs in the red flour beetle, Tribolium castaneum” aiming to intergeneration of an important roles of MicroRNAs (miRNAs) in insect growth and development, but they were poorly studied in insects. The truther analysis identified 179 differentially expressed unique miR-NAs (DEmiRNAs) during development. GO enrichment analysis indicates that the targets were enriched by protein phosphorylation, calcium ion binding, sequence-specific DNA binding transcription factor activity, and cytoplasm. They were shown to have critical roles in embryogenesis, growth, and metamorphosis of insect. This study has completed the identification and characterization of development related miRNAs in T. castaneum, and will enable researcher to investigate the role of miRNAs in insect growth and development. Some details must be revised, mainly on references, but the work is supported by the results and the proper literature. In this sense, the manuscript can be accepted in Frontiers in Environmental Science after minor revisions.

Basic reporting:

The manuscript is well written, technically correct, and follows the expected article structure. The literature references and background are correct and results obtained are relevant for the research field with an important contribution role of miRNAs in insect growth and development

Experimental design:

The background provides support for the research question evaluated and the experimental design is correct

Validity of the findings:

The results are relevant and obtained according to the expected experimental design with accuracy, supporting the conclusions. Findings of this research suggest that These miRNAs play critical roles in embryogenesis, growth, and metamorphosis of insect. This study completes the identification of development related miRNAs in T. castaneum

Minor considerations

1.     Line: 19-20: Please fuse the paragraphs “They were shown to have critical roles in 19 embryogenesis, growth, and metamorphosis of insect.”.

2.     Please add more detail regarding to the function of MicroRNAs in introduction section.

3.     Updated references should be used in Introduction.

4.     Line: 175: Figure 1and 2. Please add minor ticks on Y-axis.

5.     Please use black font color of Figure 3.

6.     Please increase the font size of Figure 4

7.     Line: 168. The involvement of these miRNAs in development will be discussed in the following section

8.     Line: 220The insects were reared in whole wheat flour containing 5% yeast at 30 °C. please revise this sentence

9.     Generally, the quality of sentences maybe uniform and professional around the MS. The typing error and grammar throughout the MS should be corrected.

10.  The conclusion section should be concise and explanatory. 

Author Response

In this work, the authors proposed an interesting study ‘” Identification and characterization of development related mi-croRNAs in the red flour beetle, Tribolium castaneum” aiming to intergeneration of an important roles of MicroRNAs (miRNAs) in insect growth and development, but they were poorly studied in insects. The truther analysis identified 179 differentially expressed unique miR-NAs (DEmiRNAs) during development. GO enrichment analysis indicates that the targets were enriched by protein phosphorylation, calcium ion binding, sequence-specific DNA binding transcription factor activity, and cytoplasm. They were shown to have critical roles in embryogenesis, growth, and metamorphosis of insect. This study has completed the identification and characterization of development related miRNAs in T. castaneum, and will enable researcher to investigate the role of miRNAs in insect growth and development. Some details must be revised, mainly on references, but the work is supported by the results and the proper literature. In this sense, the manuscript can be accepted in Frontiers in Environmental Science after minor revisions.

Basic reporting:

The manuscript is well written, technically correct, and follows the expected article structure. The literature references and background are correct and results obtained are relevant for the research field with an important contribution role of miRNAs in insect growth and development

Experimental design:

The background provides support for the research question evaluated and the experimental design is correct

Validity of the findings:

The results are relevant and obtained according to the expected experimental design with accuracy, supporting the conclusions. Findings of this research suggest that These miRNAs play critical roles in embryogenesis, growth, and metamorphosis of insect. This study completes the identification of development related miRNAs in T. castaneum

Minor considerations

  1. Line: 19-20: Please fuse the paragraphs“They were shown to have critical roles in 19 embryogenesis, growth, and metamorphosis of insect.”.

Response: This sentence has been replaced by “Knocking down the DEmiRNAs tca-miR-6-3p, tca-miR-9a-3p, tca-miR-9d-3p, tca-miR-11-3p, and tca-miR-13a-3p led to defects in metamorphosis and wing development.” in the revised manuscript.

  1. Please add more detail regarding to the function of MicroRNAs in introduction section.

Response: We made change as suggested and listed more functions of miRNAs in the section of introduction of revised manuscript.

  1. Updated references should be used in Introduction.

Response: We accept the suggestion, and added many latest articles into the section of introduction of revised manuscript.

  1. Line: 175: Figure 1and 2. Please add minor ticks on Y-axis.

Response: We have made revision on Figures 1and 2.

  1. Please use black font color of Figure 3.

Response: We made revision as suggested.

  1. Please increase the font size of Figure 4

Response: We made change as suggested.

  1. Line: 168. The involvement of these miRNAs in development will be discussed in the following section

Response: This sentence has been deleted from the revised manuscript.

  1. Line: 220The insects were reared in whole wheat flour containing 5% yeast at 30 °C. please revise this sentence

Response: This sentence has been changed as follows “The Georgia-1 (GA-1) strain of T. castaneum was reared at 30 °C in 5% yeast flour” in the revised manuscript.

  1. Generally, the quality of sentences maybe uniform and professional around the MS. The typing error and grammar throughout the MS should be corrected.

Response: We have edited the language throughout the manuscript.

  1. The conclusion section should be concise and explanatory. 

Response: We accept the suggestion and the conclusion has been changed as follows “In conclusion, we identified 179 DEmiRNAs among five developmental stages of T. castaneum and found 102 DEmiRNAs specifically expressed in EE, LL, EP and EA. Knockdown of five DEmiRNAs caused defects in metamorphosis and wing development. This study completed the identification of development-related miRNAs in T. castaneum.” (lines 267-271) in the revised manuscript.

Round 2

Reviewer 1 Report

The authors have made an important effort to improve their Ms specially by validating some of the microRNAs identified in the screening. Nevertheless, this validation is still incomplete. Besides the phenotype analysis, it is important to demonstrate that those microRNAs downregulated, at least one of the predicted gene targets that could account for the phenotype obtained upon depletion. In addition, it is important to understand the logic of the experiments, the authors need to explain the reason why they have chosen this bunch of candidates. Are they representative of the different developmental stages, that they used to classified the microRNAs?

Author Response

Thank you very much for your useful comments and suggestions. We admit that our validation of miRNAs is primary. The purpose of this study is identifying development-related miRNAs, and our published articles have validated the functions of miRNAs (tca-miR-8-3p and tca-miR-3017b) and their targets in Tribolium castaneum [1, 2]. We will verify the functions of miRNAs and their targets in the future study. Five miRNAs (three from cluster A, and one each of cluster C and E) that were selected for functional analysis are because they were shown by GO analysis to have critical roles in development of insect, and we have indicated this on lines 192-195 of the revised manuscript.

  1. Tang, J.; Zhai, M.; Yu, R.; Song, X.; Feng, F.; Gao, H.; Li, B., MiR-3017b contributes to metamorphosis by targeting sarco/endoplasmic reticulum Ca(2+) ATPase in Tribolium castaneum. Insect molecular biology 2022, 31, (3), 286-296.
  2. Wu, W.; Zhai, M.; Li, C.; Yu, X.; Song, X.; Gao, S.; Li, B., Multiple functions of miR-8-3p in the development and metamorphosis of the red flour beetle, Tribolium castaneum. Insect molecular biology 2019, 28, (2), 208-221.

Reviewer 2 Report

This manuscript can be accepted for publication.

Author Response

Thank you very much for your useful comments and suggestions.

Round 3

Reviewer 1 Report

I would like to thank the authors for addressing my concerts about the validation of the hits. I think now this section is clear enough, and tehrefore I recommend this manuscript for publication.